# In-silico drug repurposing study predicts the combination of pirfenidone and melatonin as a promising candidate therapy to reduce SARS-CoV-2 infection progression and respiratory distress caused by cytokine storm

**Laura Artigas[1], Mireia Coma[1]\*, Pedro Matos-Filipe[1,2], Joaquim Aguirre-Plans[2], Judith Farrés[1], Raquel Valls[1], Narcis Fernandez-Fuentes[3], Juan de la Haba-Rodriguez[4], Alex Olvera[5], Jose Barbera[6], Rafael Morales[6], Baldo Oliva[2]\*, Jose Manuel Mas[1]**

1 Anaxomics Biotech, Barcelona, Spain, 2 Structural Bioinformatics Group, Research Programme on Biomedical Informatics, Department of Experimental and Health Science, Universitat Pompeu Fabra, Barcelona, Catalonia, Spain, 3 Department of Biosciences, U Science Tech, Universitat de Vic—Universitat Central de Catalunya, Vic, Catalonia, Spain, 4 Maimonides Biomedical Research Institute, Reina Sofia Hospital, University of Cordoba, Cordoba, Spain, 5 Institut de Recerca de la Sida—IrsiCaixa, Hospital Universitari Germans Trias i Pujol, Badalona (Barcelona), Spain, 6 Servicio de Medicina interna—Unidad de Infecciosas, La Mancha—Centro Hospital, Alcázar de San Juan, Spain

\* mcoma@anaxomics.com (MC); baldo.oliva@upf.edu (BO)

## Abstract

From January 2020, COVID-19 is spreading around the world producing serious respiratory symptoms in infected patients that in some cases can be complicated by the severe acute respiratory syndrome, sepsis and septic shock, multiorgan failure, including acute kidney injury and cardiac injury. Cost and time efficient approaches to reduce the burthen of the disease are needed. To find potential COVID-19 treatments among the whole arsenal of existing drugs, we combined system biology and artificial intelligence-based approaches. The drug combination of pirfenidone and melatonin has been identified as a candidate treatment that may contribute to reduce the virus infection. Starting from different drug targets the effect of the drugs converges on human proteins with a known role in SARS-CoV-2 infection cycle. Simultaneously, GUILDify v2.0 web server has been used as an alternative method to corroborate the effect of pirfenidone and melatonin against the infection of SARS-CoV-2. We have also predicted a potential therapeutic effect of the drug combination over the respiratory associated pathology, thus tackling at the same time two important issues in COVID-19. These evidences, together with the fact that from a medical point of view both drugs are considered safe and can be combined with the current standard of care treatments for COVID-19 makes this combination very attractive for treating patients at stage II, non-severe symptomatic patients with the presence of virus and those patients who are at risk of developing severe pulmonary complications.

**Data Availability Statement:** All relevant data are within the manuscript and its Supporting Information files.

**Funding:** LA, MC, PMF, JF, RV and JMM have commercial affiliation to Anaxomics Biotech SL. The funders provided support in the form of salaries for authors JAP, NFF and PMF, but did not have any additional role in the study design, data collection and analysis, decision to publish, or preparation of the manuscript. The specific roles of these authors are articulated in the 'author contributions' section. JAP, NFF and BO acknowledge support from the Spanish Ministry of Economy (MINECO) [BIO2017-85329-R] [RYC-2015-17519]; Unidad de Excelencia María de Maeztu", funded by the Spanish Ministry of Economy [ref: MDM-2014-0370].PMF has received funding from the European Union's Horizon 2020 research and innovation programme under the Marie Skłodowska-Curie grant agreement No 765912.

**Competing interests:** The commercial affiliation of the authors to Anaxomics Biotech SL. does not alter our adherence to PLOS ONE policies on sharing data and materials

**Abbreviations:** ANN, Artificial Neural Networks; ARD, Acute Respiratory Distress; SARS, Severe Acute Respiratory Syndrome; EMA, European Medicines Agency; GEO, Gene Expression Omnibus; MoA, Mechanism of Action; TPMS, Therapeutic Performance Mapping System.

## Introduction

The pandemic disease COVID-19 caused by SARS-CoV-2 is an active infection that has affected at least hundreds of thousands of individuals around the world. The asymptomatic incubation period varies greatly among patients ranging from 2 to 14 days and up to 27 days for some reported cases [1]. COVID-19 typically causes flu-like symptoms including fever and cough. Within a week, 20% of the patients develop respiratory complications such as pneumonia, with chest tightness, chest pain, and shortness of breath [2].

Currently, no specific treatment has been approved for COVID-19. Standard treatments focus on the symptoms, such as pain relievers (paracetamol or acetaminophen) and breathing support (mechanical ventilation) for patients with severe infection. Other repurposed therapies aiming at reducing viral load, spread or protecting the host machinery are currently being tested experimentally in more than 600 clinical trials, involving medicines such as: human immunoglobulin, interferons, chloroquine, arbidol, remdesivir, favipiravir, lopinavir, ritonavir, oseltamivir, methylprednisolone, bevacizumab, and traditional Chinese medicines [3]. Out of them, remdesivir, has shown encouraging results in a clinical trial, pointing that it is possible to block the virus using drugs [4].

Drug repurposing (i.e. reusing existing drugs approved for different conditions and with acceptable safety profiles) is emerging as a rapid and excellent cost-benefit ratio alternative to find treatments against SARS-CoV-2, while vaccines and new treatments are being developed. However, there are two key aspects to be considered on the search of drug repurposing candidates. First of all, we require to identify the key proteins or pathways that the treatment should target. These targets can either be proteins of the virus involved in the mechanisms used to enter the host cell and replicate, or the host proteins that facilitate the spread of the virus or cause over-reactive dangerous responses. Proteomics studies such as the works of Gordon et al. [5], Zhou et al. [6] or included in databases such as IntAct [7] uncover the target elements of the human interactome. Second, we require to find the treatments that target the key proteins and pathways identified, to either inhibit or activate them in order to undermine the viral load or improve the host response.

All proposed treatments so far are repurposed drugs selected by the similarity of their original indication, like antivirals and antiretrovirals, or the similarity in their mechanism of action. But the arsenal of existing drugs can still be further exploited by identifying new mechanisms of action and exploring combinatory effects. The most efficient way of evaluating these possibilities is to apply computational methods and, more specifically, systems biology and artificial intelligence-based methods [6, 8–10]. Systems biology methods do not focus on the effect of drugs on single biological entities, instead they evaluate their effect on systems of interconnected biological entities. By exploiting the knowledge on poly-pharmacology together with the high connectivity among cellular processes, systems biology methods are able to identify new therapeutic uses for approved drugs [11].

The objective of this work is to identify existing drugs, or a combinations of them, using a systems biology and artificial intelligence-based approach, the Therapeutic Performance Mapping System (TPMS) technology [12], that could help reducing both the infection capacity of the virus and its associated complications. The TPMS technology is a top-down approach which mathematically models the human physiology by integrating the known information about the functional interaction of proteins with experimental and patient data available in public repositories, with a focus on clinical translation. There are already several examples of repurposed drugs identified using TPMS approach that have been validated in vitro and in vivo [8, 9, 12–18] with one of them advancing to clinical trials. Simultaneously, GUILDify v2.0 web server [19] has been used as an alternative method to corroborate the promising combinations.

## Methods

### Identification of host key-points of intervention

The identification of host key points of intervention has been done through manual curation of scientific publications and gene expression analysis of data included in the Gene Expression Omnibus (GEO) database [20]. This has provided 3 sets of proteins related with the infection process: 1) coronavirus-host interaction set (including SARS-CoV-2 entry points), 2) lung-cells infection set, and 3) acute respiratory distress (ARD) set.

The 'Coronavirus-host interaction set' is composed of human proteins with a relevant role in SARS-CoV-2 infection and a set of human coronaviruses host interactome retrieved through manual curation of scientific publications (S1 Table).

The 'lung-cells infection set' has been populated through differential expression of the GSE147507 [21] gene expression dataset using edgeR [22] package (p-value <0,05 and log2FC>1 or <-1).

The 'ARD set' has been defined using microarray data sets included in GEO that define the role of different important cell types in ARD pathophysiology: GSE89953 (alveolar macrophages and monocytes ARD) [23], GSE76293 (Neutrophils in ARD) [24] and GSE18712 (lung cells at different ARD stages) [25]. The first 2 sets have been analysed with the Geo2R tool [20], and the last one with a statistical analysis method based on Significance Analysis of Microarrays (SAM) [26, 27] to determine the differentially expressed proteins. Additionally, ARD cytokine storm has been characterized by manual curation of scientific literature (S1 Table).

### TPMS technology: Systems biology analysis for drug-repurposing discovery

The TPMS technology employed has been previously described [12] and applied in different clinical areas with different objectives [8, 9, 13–18]. TPMS uses a human biological network that incorporates the available relationships (edges or links) between proteins (nodes) from a regularly updated in-house database drawn from public sources: KEGG [28], REACTOME [29], INTACT [30], BIOGRID [31], HPRD [32], and TRRUST [33].

Drug targets and indications were obtained from DrugBank [34]. The molecular description of the indications was obtained from a hand-curated collection of associations between biological processes and molecular effectors (defined as BED, Biological Effectors Database, from Anaxomics Biotech). The method uses an artificial neural network (ANN) to measure the potential relationship between the nodes of a network (i.e. proteins), grouped according to their association with a phenotype. The ANN algorithm provides a score (from 0 to 100%). Each score is associated with a probability (p-value) that the protein or group of proteins being evaluated, drug targets with molecular description of pathological processes, are functionally associated. Scores greater than 91% indicate a very strong relationship with a p-value below 0.01; scores between 76–91% have p-value between 0.01–0.05; scores between 40–76% have medium–strong relationship and p-value in the range 0.05–0.25; and a scores lower than 40% have p-values above 0.25.

### TPMS technology: Systems biology analysis for mechanism of action discovery

The mechanism of action (MoA) of pirfenidone and melatonin has been simulated by using the TPMS technology. On the basis of the human protein functional network described above, mathematical models have been built. The models simulate the activation/inhibition status in the human protein network. As input, the model takes the activation (+1) or inactivation (-1)

of the drug target proteins (S2 Table), and as output the protein states of the pathology of interest (S1 Table). It then optimizes the paths between the two protein sets and computes the activation and inactivation values of the full human interactome. The resulting subnetwork of proteins with positive and negative values defines the MoA of the drug. Sampling methods are used to generate mathematical models with stratified ensembles that comply with a set of validated benchmarks [12].

### GUILDify v2.0 web server to calculate the effect of the COVID-19 sets of proteins and repurposed drugs to the network

GUILDify v2.0 [19] web server is used to extend the information of sets of proteins through the human biological network. GUILDify scores proteins according to their proximity with the genes associated with a drug or phenotype (seeds). In this study, GUILDify has been employed to calculate the neighbourhoods of the human biological network that are associated with the host-key points (for SARS-CoV infection) and at the same time affected by specific drugs. In this way, the mechanism of action of the treatments proposed by TPMS can be corroborated using a completely different setting.

GUILDify v2.0 has been run for each drug, using their targets as seeds (by introducing the name of the drug in the search box). The same has been done for each set of proteins: the coronavirus-host interaction set, the entry points set (a subset of the coronavirus-host interaction set containing only 5 host proteins relevant in the infection of SARS-CoV-2), the lung cells infection set and the ARD set (by uploading a file with the gene symbols of the proteins). We included the entry points subset in the analysis to have a more specific neighborhood of the SARS-CoV-2 infection mechanism. There are two additional parameters to be chosen by the user in GUILDify web server: the human biological network and the scoring function. The human biological network used has been the protein-protein interactions network of I2D [35], as it is the more complete network in the web server (with 13,568 proteins and 224,675 interactions). The scoring function selected (used to score the proteins of the network according to the proximity with the seeds) has been NetCombo [36], an algorithm that combines the performance of three network-based prioritization algorithms: NetShort (considers the number of edges connected to phenotype-associated nodes), NetZcore (iteratively assesses the relevance of a node for a given phenotype by averaging the normalized scores of the neighbors) and Net-Score (considers the multiple shortest paths from the source of information to the target). The top 1% scored proteins of each test have been selected to proceed with the analysis of each subnetwork. The overlap between the selected subnetworks has been used to calculate the number of common proteins between the top-scoring proteins associated with the drugs and the top-scoring proteins associated to the host-key points (results detailed in S3 Table).

## Results and discussion

### Identification of host key-points for intervention

Human key molecular enclaves in COVID-19 that could be good target areas for treatment have been identified through manual curation of scientific publications and gene expression analysis, describing molecularly aspects of the virus infection process from the host perspective and aspects related with the respiratory problems associated to the infection. We have defined three different main protein sets: 1) coronavirus-host interaction set, 2) lung cells infection set and 3) ARD set.

1) 'Coronavirus-host interaction set' is composed of human proteins with a relevant role in SARS-CoV-2 infection process. This set of proteins has been obtained from a bibliographic

revision and it is composed of proteins that potentially interact with SARS-CoV-2: TMPRSS2 [37], ACE2 [37], GRP78 [38], Furin [39], CD147 [40]. In addition, a list of proteins reported to have a direct interaction with coronaviruses, retrieved from Zhou et al. [6] has been included. In this study, the authors define a human coronavirus (HCoV)-host interactome network including 119 host proteins with the aim of identifying antiviral drugs through repurposing strategies. It integrates known human direct targets of HCoV proteins or that are involved in crucial pathways of HCoV infection using the available information from four known HCoVs (SARS-CoV, MERS-CoV, HCoV-229E, and HCoV-NL63), one mouse CoV (MHV), and one avian CoV (IBV, N protein). The final set contains 122 human proteins, the full list is provided in S1 Table.

2) 'Lung cells infection set' describes the transcriptional response of human lung epithelial cells to SARS-CoV-2 infection. It has been defined through differential expression analysis of human lung epithelial cells to SARS-CoV-2 infection, GEO series reference GSE147507. This set includes 47 proteins, the full list is provided in S1 Table.

3) 'ARD set' is composed of 412 unique protein entries defined by 6 subsets. It includes gene differential expression data on alveolar macrophages, monocytes and neutrophils in ARD (GSE89953 and GSE76293), gene differential expression data on lung changes induced in the intermediate and late stages of the pathophysiology (GSE18712) and the key players in ARD cytokine storm extracted from literature research. All ARD set data can be retrieved from S1 Table.

We have centred the drug repositioning analysis in the coronavirus-host interaction set. But the potential effects of the drug combination selected have been evaluated for the 3 data sets defined. An overall depiction of the whole procedure is depicted in Fig 1. TPMS-ANN approach has been used to identify drugs that can be repurposed for set 1. The potential MoA for these repurposed drugs has been identified by GUIILDify and TPMS approaches, as well as MoAs of this repurposed drug candidates for sets 2 and 3.

## Individual drug repurposing analysis

A systems biology based strategy has been used to screen the predicted relationship between 6,605 different drugs, as defined in Drugbank [34], and the coronavirus-host interaction set previously defined. Drug candidates for repurposing were sorted and selected by the probability of its relationship with the coronavirus-host interaction set, 12 approved drugs scored higher than 74% (p-value <0.05). All candidates were grouped in two drug categories: (1) Anti-inflammatory and immunosuppressant agents (ibuprofen, phosphatidyl serine, prasterone, vitamin C, pirfenidone, tacrolimus and pimecrolimus); (2) Cardiovascular agents (isosorbide, icatibant, moexipril, irbesartan, phenindione). Among them, 4 out of 12, are currently being tested in COVID-19 clinical trials, these are: vitamin C (NCT04264533, NCT04323514, NCT04323514), pirfenidone (NCT04282902), tacrolimus (NCT04341038) and irbesartan (NCT04330300). Two others appear described as potential treatments by third authors, moexipril [41] and icatibant [42], or somehow related to the disease outcome, such as ibuprofen [43].

Among these candidates, pirfenidone (with score 75%) was selected because of its reported association with furin [39] and a good safety profile. The S protein of SARS-CoV-2 has a furin-like domain for cleavage and some recent works have addressed the potential implication of this domain in the S protein maturation and, as a consequence, in the host cell entry mechanism of the virus. Furin, a member of the subtilisin-like proprotein convertase family processes proteins of the secretory pathway, is a type 1 membrane-bound protease that is expressed in multiple organs, including the lungs [44]. SARS-CoV-2, differently from SARS-CoV, presents

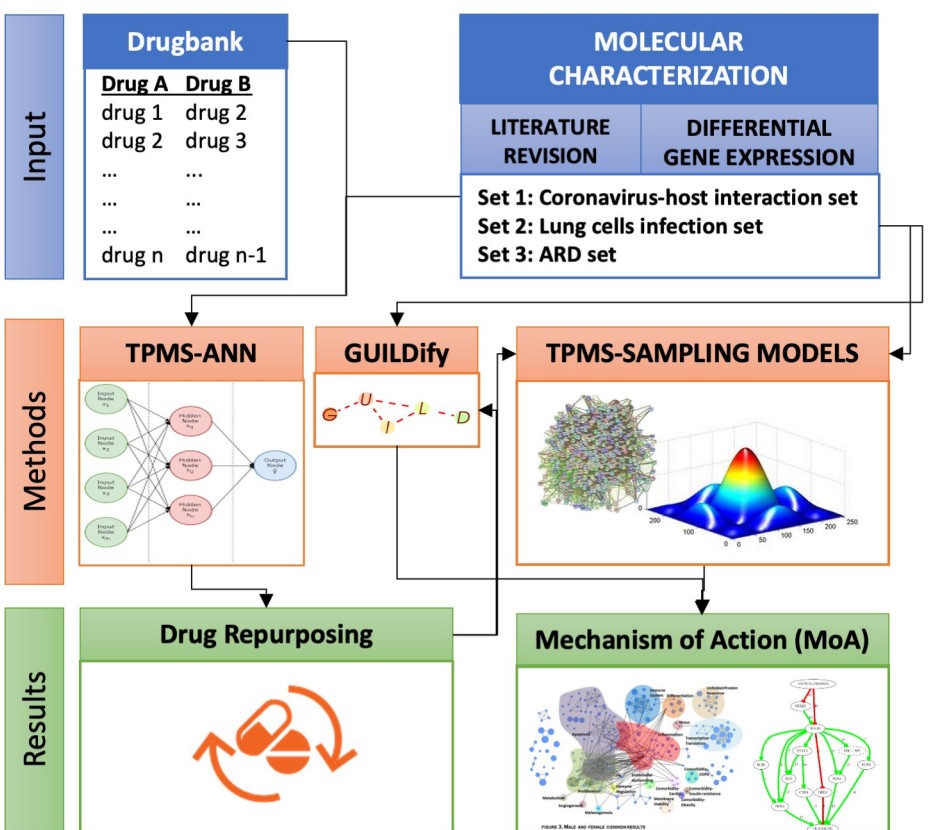

**Fig 1. Overall approach.** TPMS-ANN approach has been used to identify drugs for repurposing against set 1. The potential MoA of drugs repurposed for sets 1, 2 and 3 have been identified by TPMS and GUILDify approaches.

a furin cleavage site between the S1/S2 subunits of the S glycoprotein, which is cleaved during the biogenesis of the virus to create a mature S protein [45]. Inhibiting the expression of furin could then be a possible approach to diminish SARS-CoV-2 infectivity by making difficult S protein maturation [46].

In recent years, furin has emerged as a promising target for therapeutic intervention in a variety of infectious diseases as influenza A virus [47] or Newcastle disease virus [48]. Given this evidence, pirfenidone stands out as one of the most interesting repurposing candidates to treat COVID-19. Taking into consideration the drug repurposing analysis and the supporting evidence we have selected pirfenidone for further studies.

## Drug combination repurposing analysis

Combining different drugs can be an advantageous option as we can affect a larger number of key points of intervention and/or increase the effects against disease pathophysiology. We have applied ANN of TPMS technology to screen drug combinations composed by pirfenidone and other approved drugs listed in Drugbank database. Among all, 214 non-synonymous repurposed drug candidates have shown a predicted relationship value $\geq$ 90%. Experimental drugs, withdrawn drugs and those with poor safety profile or with opposite mechanistic effect than the desired one were further filtered out. Lastly, we selected the combination of pirfenidone (predicted value 92% p>0.05) with melatonin as it fulfilled all criteria. Furthermore, there is also mounting evidence that melatonin could be a good candidate to treat COVID-19,

due to its anti-inflammatory and anti-oxidative properties protecting against ARDS caused by other pathogens including viruses [6, 49–51].

## Mechanism of action of pirfenidone-melatonin combination against the virus-host interaction network

To further characterize at a molecular level the action of the selected drug combination on the coronavirus-host interaction set, we have used the modelling strategy based in sampling methods of TPMS. The models predict the most probable molecular routes that transmit the effect from the drugs' molecular targets to proteins in coronavirus-host interaction set.

The main protein interactors identified are depicted in Fig 2. Pirfenidone inhibits furin, a protein effector included in the virus-host network, with a potential implication for the entry of the virus in the host cell. The inhibition of furin also results in the inhibition of transforming growth factor-beta1 (TGF-β1) [52] a gatekeeper of the immune response and one of the key

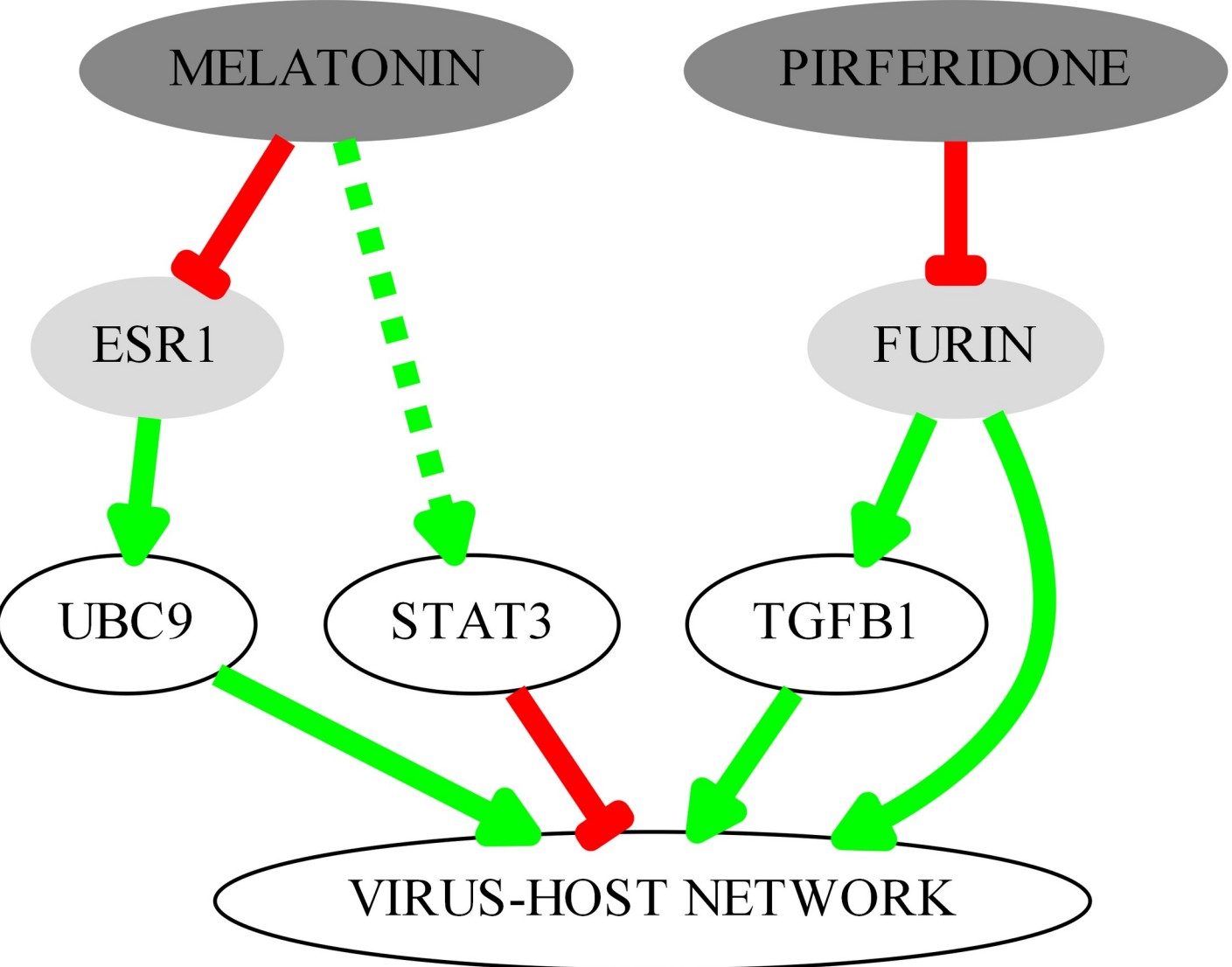

**Fig 2. Key interactors in the predicted mechanism of action of pirfenidone and melatonin combination therapy targeting the coronavirus-host interaction set.**

proteins in Set 1, as it is upregulated by the papain-like protease (PLpro) from Severe acute respiratory syndrome (SARS) coronavirus (SARS-CoV) [53]. The inhibition of furin (a TGF-β1 converting enzyme) by pirfenidone reduces its expression [52, 54]. On the other hand, melatonin, through the suppression of estrogen receptor alpha [55] regulates the expression of UBC9 [56], another key protein of set 1 that interacts with SARS-CoV N protein [57]. It has also been described that melatonin can lead to the activation of STAT3 in some cell types [58, 59], and it could be useful to counteract the STAT3 dephosphorylation induced by SARS-CoV [60].

## GUILDify v2.0 to confirm the effect of pirfenidone and melatonin towards the SARS-CoV-2 infection cycle set

In the previous section, the drug combination pirfenidone and melatonin was selected as a potential treatment against the infection of SARS-CoV-2. The selection is fundamentally based on the mechanism of action of the combination, targeting the entry mechanism of the virus and specially furin, a protein that may have a key role in the infection process [39]. To corroborate the effect of pirfenidone and melatonin to the proteins that are relevant for the infection, we have used an alternative approach also based on the use of networks. We used GUILDify v2.0 [19], a web server that extends the information of sets of proteins through the human biological network. GUILDify scores proteins according to their proximity with the genes associated with a drug or phenotype (seeds). Using this web server, we can identify a list of top-scoring proteins that are critical on transmitting the perturbation of certain proteins through the network (also known as network propagation). Thus, applied to this case, we: (i) identify the proteins of the human biological network that are key on transmitting the therapeutic effect of pirfenidone and melatonin, (ii) identify the proteins of the network perturbed by the infection of the SARS-CoV-2, and (iii) calculate the network overlaps between the treatment and the infection, reflecting the potential effect of the treatment over the infection. The network used by GUILDify, obtained from I2D [35], is completely independent from the network used in the TPMS, becoming an ideal, independent context to test the results of TPMS.

We have observed a significant overlap of 32 proteins between the top-scoring proteins achieved with the targets of pirfenidone (pirfenidone-network) and the top-scoring proteins of the SARS-CoV-2 infection cycle set (a subset of the coronavirus-interaction set specific for SARS-CoV-2, conformed by TMPRSS2, ACE2, GRP78, furin and CD147). Additionally, there are 10 common proteins between the top-scoring proteins achieved with the targets of melatonin (melatonin-network) and the top-scoring proteins of the 5 entry point proteins. There are also 7 common proteins between the top-scoring proteins of pirfenidone and the top-scoring proteins of ARD set. However, the rest of the sets (coronavirus-host interaction set and lung infection set) do not have a significant overlap with any of the two drugs (results detailed in S3 Table).

The highly significant overlap between pirfenidone and the top-scoring proteins of the 5 entry point proteins of SARS-CoV-2 is however biased, because furin is in this set and it's also a target of pirfenidone (Fig 3). The therapeutic effect of pirfenidone on furin has a clear source effect on the neighbourhood of proteins that surround the entry points of SARS-CoV-2. Melatonin has also a significant overlap with the entry points of SARS-CoV-2. This effect is explained because it targets the proteins MTNR1A and MTNR1B (melatonin receptors). Melatonin affects SARS-CoV-2 GRP-78 entry point (also called HSPA5) because MTNR1A and MTNR1B are direct interactors of GRP-78 (Fig 3). Melatonin also affects SARS-CoV-2 furin entry point because MTNR1B interacts with the membrane protein ITM2C, which directly interacts with furin. By identifying the interacting partners of MTNR1A and MTNR1B, we can

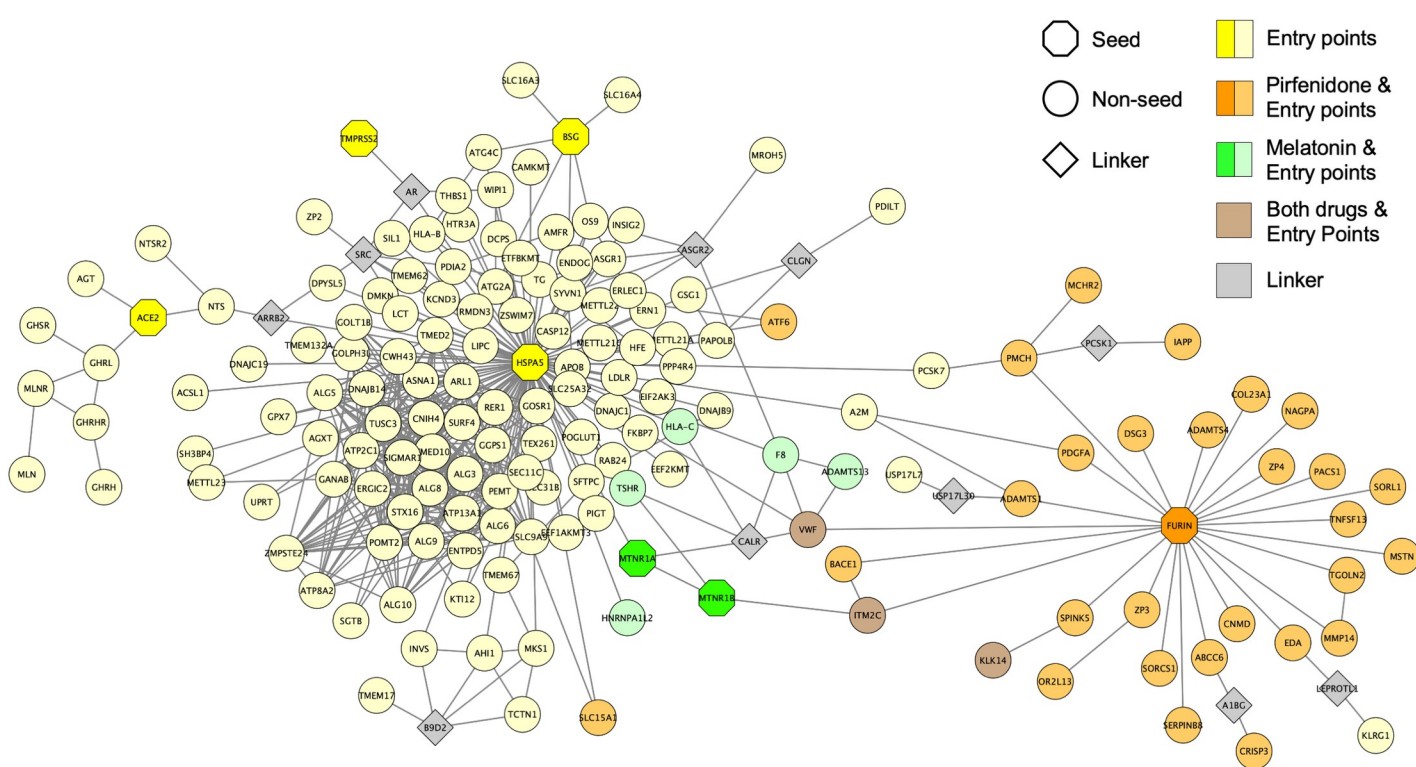

**Fig 3. Network of the SARS-CoV-2 entry points and the shared interactions with pirfenidone and melatonin.** The extended set of proteins associated with the SARS-CoV-2 entry points, pirfenidone and melatonin were obtained with GUILDify v2.0. In the network, seeds are represented as octagons, non-seeds as circles and "linkers" as diamonds. "Linkers" are non-top-scoring proteins (they do not belong to any of the represented sets) that link unconnected groups of proteins to the largest connected component. "Linkers" are represented in this plot to facilitate the visualization, but they are not associated to any of the sets of proteins, thus having a less relevant role in the interpretation of the results. Pirfenidone-associated proteins are coloured in orange, melatonin-associated proteins are coloured in green and proteins associated with both drugs in brown.

understand better the potential mechanism of action of melatonin towards the entry of the virus. Apparently, both pirfenidone and melatonin target close subnetworks but in different areas. This could explain the interesting additive effect predicted by TPMS. As mentioned above, the overlap between pirfenidone-network and the SARS-CoV-2 entry points is biased by the fact that both sets of proteins contain furin. Therefore, we have repeated the same analysis but removing furin from the set of SARS-CoV-2 entry points. Here, we have observed that the overlap with pirfenidone is not significant, with only 4 common proteins, which proves that pirfenidone targets the SARS-CoV-2 network specifically by targeting furin. In contrast, the melatonin-network has a significant overlap of 7 proteins, affecting mainly neighbours of GRP-78. GUILDify also reflects the effect of pirfenidone over 7 proteins related with ARD, leaded by other targets of the drug: TNF-alpha and MAPK13. According to the findings by GUILDify, we confirm the effect of the combination of pirfenidone and melatonin in the entry points of the SARS-CoV-2 infection, specifically the neighbours of furin and GRP-78, and some proteins associated with ARD.

### Evaluation of pirfenidone and melatonin combination over respiratory distress

To further evaluate the impact of pirfenidone and melatonin as a novel therapy for COVID-19, the relationship of the drug combination against the associated respiratory problems has been evaluated using various systems biology approaches.

In a first approach, the relationship between the protein drug targets and the 2 sets of proteins defining respiratory problems associated with COVID-19 (Sets 2 and 3) has been evaluated using the TPMS ANN modelling approach. This type of measurement provides a probability of a relationship between groups of proteins. A weak relationship has been obtained for the combination with lung cells infection set while we obtained a strong relationship between the drug combination and ARD set. Specifically, with the subsets related to alveolar macrophages, monocytes, late phase ARD and ARD cytokine storm.

A second approach using the sampling methods-based models of TPMS has been used to evaluate at the molecular level the MoA of pirfenidone and melatonin individually and in combination for treating ARD set, evaluating each subset individually. This modelling approach identifies a probable molecular pathway between the protein drug targets and the ARD subsets. The ability of each treatment to reverse the protein alterations occurring in these pathological mechanisms has been assessed, i.e. the ability of each drug to activate proteins that are inhibited in ARD subnetworks or vice versa (according to the molecular characterization). We have calculated the percentage of proteins theoretically affected respect to the total number defining the subset (Fig 4). Fig 4 displays the % and absolute value of affected proteins predicted by the MoA by the individual drugs and by the drug combination for each of the ARD subsets. Last column provides the overlap between the proteins affected in each of the individual MoAs. The detail of the proteins affected is listed alongside the list of Set 3 proteins in S1 Table. The main effect, for both drugs, is over the cytokine storm subset. Pirfenidone and melatonin are able to affect 86% of the proteins defining the set individually. The drug combination increases the % of proteins affected to a 91%, providing a potential synergy in this area. The protein-network overlap indicates that most of the proteins (82%) in ARD cytokine storm subset are modulated by both drugs. While the proteins modulated by one drug alone are 5% in both cases. The second subset with largest % of affected proteins, 27% for the drug combination, is alveolar macrophages set. The main effect is registered for melatonin, pirfenidone contribution is negligible for this subset. We measure also a minor modulation for the rest of ARD subsets, mostly contributed by melatonin with pirfenidone slightly potentiating the effect in some of them.

The anti-inflammatory role of both melatonin and pirfenidone has been described in the literature. They are able to attenuate NLRP3 inflammasome and IL-1β [61–63], being a possible starting point for the cytokine storm. Studies have shown the potential of melatonin in decreasing the levels of inflammatory cytokines by reducing the activation of NF-κB, thus becoming an interesting candidate to treat different types of viral infections [49, 64, 65]. Other studies have also shown the anti-inflammatory role of pirfenidone, modulating inflammatory cytokines [66, 67]. The anti-inflammatory effect of both treatments could potentially be useful to reduce the effects caused by the cytokine storm. Pirfenidone has also been described as antifibrotic, reducing the rate of lung damage by 50% in patients with idiopathic pulmonary fibrosis [68]. Antifibrotic therapy has been highlighted as a potential strategy to treat COVID-19 patients and prevent fibrosis after SARS-CoV-2 infection [69].

In addition, both pirfenidone [70, 71] and melatonin [72, 73] have been reported to provide antioxidant effect, which has an important role in ARD pathophysiology. Multiple studies have reported reduced levels of antioxidant agents in ARD patients [74, 75], and remark the positive effects of therapies that include antioxidants in the treatment of ARD [76, 77]. Therefore, the antioxidant effect of both drugs could be one of the factors that explains the positive effect in ARD predicted by TPMS.

The proposed drug combination has also a relevant effect modulating the expression of proteins associated to the phenotype of alveolar macrophages in ARD. It has been suggested that these cells are key players in the pathophysiology of ARD. They seem to have a proinflammatory role in early stage and anti-inflammatory effect at the late stage [78].

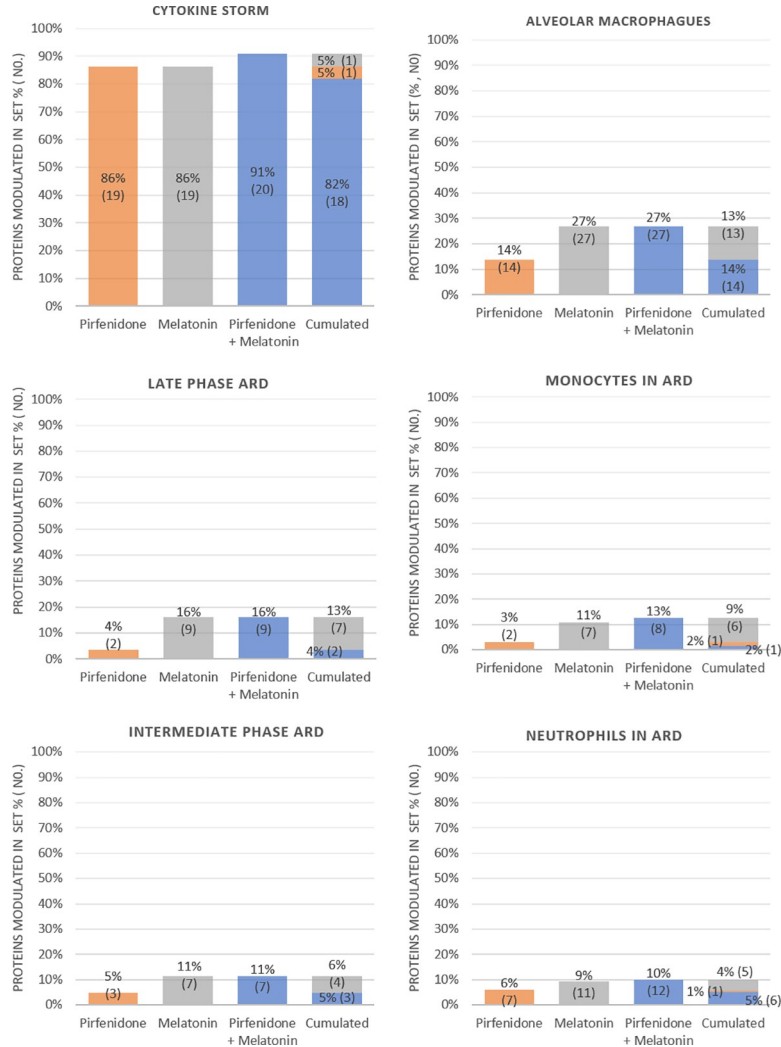

**Fig 4. Proteins associated to ARD set modulated by pirfenidone and melatonin individually and in combination.** The MoA has been calculated for each drug individually and in combination for each of the subsets defining ADR set. The first 3 columns provide the % and absolute value of affected proteins in the set for each MoA. Last column provides the cumulative, showing the protein overlaps affected between the different treatments.

## Mechanism of action of pirfenidone-melatonin combination over respiratory distress associated proteins

The combined mechanism of action (MoA) of pirfenidone and melatonin in ARD have been identified at molecular level through the use of sampling methods-based models [12]. TPMS sampling-based methods trace the most probable paths, both in biological and mathematical terms, which lead from a stimulus (e.g. drug) to a response (e.g. disease) through the biological human protein network. In other words, TPMS identifies the set of possible MoAs that achieve a physiological response when the system is stimulated with a specific stimulus. There is a certain variability in these MoAs, generating several possible pathways that represent patient heterogeneity. However, when identifying a graphical representation of the MoA, only the mean and most represented paths among the set of possible solutions is represented.

The MoA of the combination of pirfenidone and melatonin in ARD has been studied. As shown in Fig 5, and as previously predicted through other strategies, the combination of both

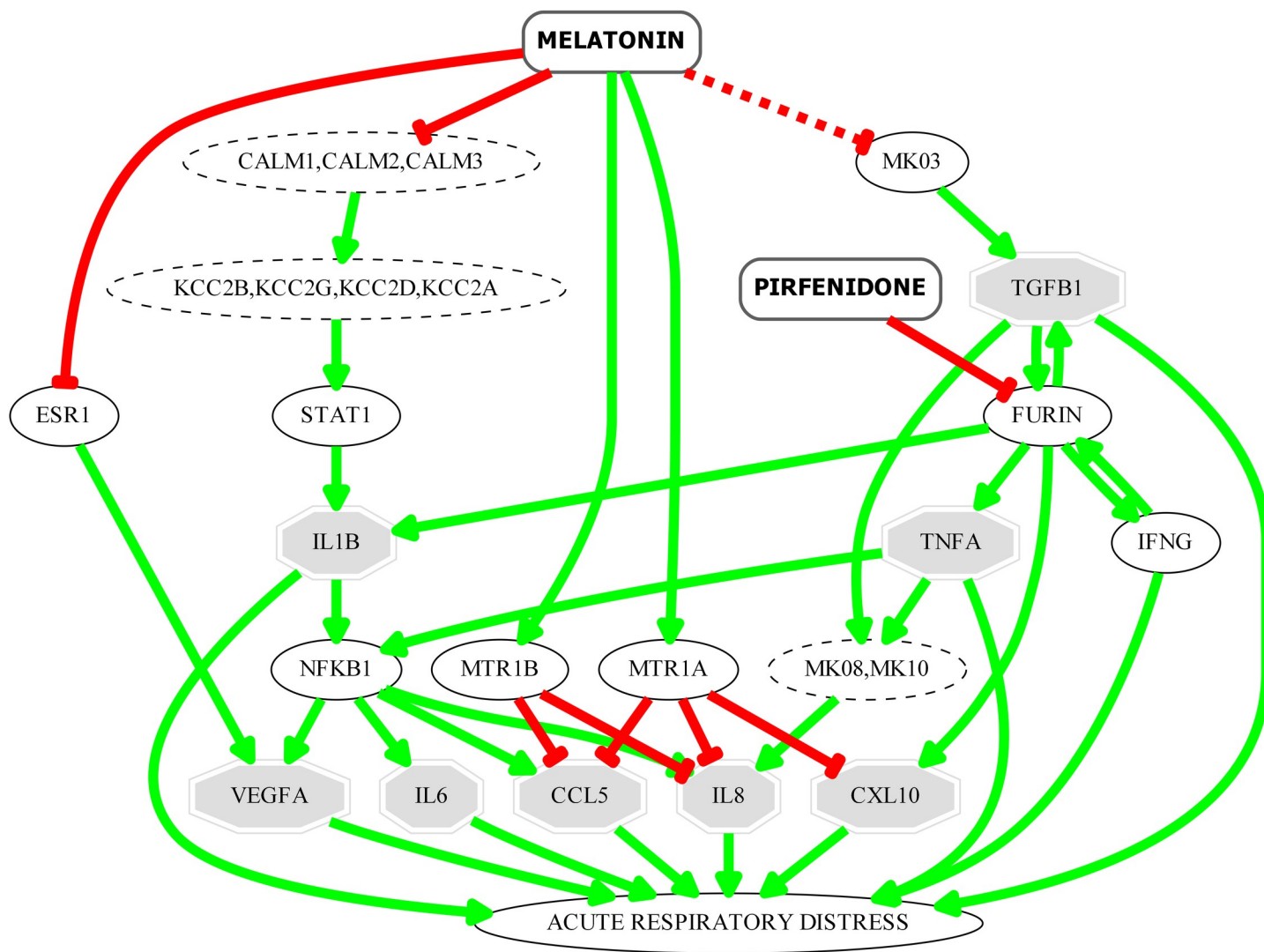

**Fig 5. Key interactors in the predicted mechanism of action of pirfenidone and melatonin combination over the ARD set.** Only the most relevant ARD effectors appear in the figure, represented as octagons.

drugs counteracts the cytokine storm, which is responsible of the disease severity. Melatonin and pirfenidone may reduce the levels of proinflammatory chemokines and cytokines that have been detected at high levels in patients with COVID-19 [79] and which are well known for their role in the cytokine storm and contribution to ARD: IL-1B, IL-6, CXCL10, IL-8, CCL5 and TNF-alpha [79–82]. These molecules are involved in both, early and late phases of cytokine storm. These effects have already been previously described in the literature for both drugs individually [49, 66, 83–86] reinforcing the potential use of the drug combination to improve patient outcomes.

This drug combination could modulate TGF-β1 and VEGFA involved in processes associated to lung fibrosis, a late complication of ARD. TGF-β1 is a profibrotic mediator known to be associated with the fibroproliferative response responsible of this disease complication [87]. Pirfenidone inhibits furin, one of the proprotein convertases that mediate the maturation/activation of TGF-β1 [88]. On the other hand, it has been described that melatonin supress TGF-

β1 expression through the inhibition of ERK phosphorylation [89, 90] supressing pulmonary fibrotic response.

One key component of the infrastructure in the lung that is damaged in ARD is the vasculature. VEGFA is involved in the excessive angiogenic response that may contribute to the initial tissue injury and drive the fibroproliferative response [87]. According to our model, melatonin could be downregulating VEGFA expression through the inhibition of oestrogen receptor alpha (ER1). It has been described that melatonin inhibits ER transactivation [55] and that VEGF expression in some cell types is regulated by 17β-oestradiol in a ER dependent process [91]. There are some experimental evidences suggesting the inhibitory role of melatonin on VEGF activity reinforcing the hypothesis of our model, although the pathway involved on this effect has not been described [92–94].

In summary, we predict that this drug combination could modulate the vast majority of effectors involved in the cytokine storm. Furthermore, as detailed in Fig 4, the drugs also act on other pathophysiological subnetworks of ARD. It has also been described in the literature the anti-oxidant effect of both, pirfenidone and melatonin [70, 72, 95]. As reactive oxygen species (ROS) play a central role in inflammatory responses and viral replication [96], such anti-oxidant effect could also be beneficial for improving the pathophysiology of ARD patients.

## Conclusion

The current situation with COVID-19 prompts for easy and fast ways to control and reduce the burthen of the disease. The systems biology-based drug repurposing screening method has proved to be a good option, identifying most of the treatments already in clinical evaluation and highlighting others that may have not got sufficient attention on their potential. Our analysis has highlighted pirfenidone as one of the best candidates for treating the SARS-CoV-2 host cell entry and melatonin as combination theraphy to increase the measured effect.

The identification of the potential MoA for this drug combination against the coronavirus-host interaction set indicates that the principal effect is registered by the inhibition of furin by pirfenidone and the suppression of estrogen receptor alpha by melatonin. The effect of the two drugs in the key proteins surrounding the SARS-CoV-2 entry mechanism, furin and GRP-78, has been corroborated with the web server GUILDify v2.0.

We have also measured a potential effect of the drug combination over ARD set. Screening in a deeper detail the potential MoA we have measured the strongest modulation over cytokine storm subset of ARD with pirfenidone and melatonin contributing in a similar manner. The other ARD subsets are not so strongly modulated and most of the effect comes from melatonin.

Several authors have suggested both compounds individually to treat COVID-19 or in combination with other compounds. Pirfenidone is already in clinical trials (NCT04282902) to treat COVID-19 associated lung injury. The fact that our study has identified also an effect on SARS-CoV-2 infection cycle and thus possibly contributing to reduce the viral load makes this drug combination very attractive in the treatment of COVID-19.

From a medical point of view both drugs are considered safe and can be combined with the current standard of care treatments for COVID-19. The pharmacokinetics of the combination therapy can be altered respect to the individual drugs due to their competition (substrate–substrate) over CYP1A2. This substrate competition can lead to the drugs remaining longer in the organism than in monotherapy, potentially increasing the known pharmacologic effects of the drugs that in turn could lead to dose-dependent adverse drug reactions. The design of any clinical trial to test this combination therapy has to take into consideration this possibility and compensate by reducing the dosage of the drugs compared to their regular use in monotherapy.

This drug combination is foreseen to tackle two important issues in COVID-19, the viral load and the pulmonary associated complications making it very appropriate for treating patients' at stage II [97], this are non-severe symptomatic patients with the presence of virus and at risk of evolving to severe pulmonary complications. A clinical study to evaluate this drug combination is foreseen.

## Supporting information

**S1 Table. Characterization of the different sets of proteins associated to COVID-19.** 1) coronavirus-host interaction set (including SARS-CoV-2 entry points), 2) lung-cells infection set, and 3) acute respiratory distress (ARD) set that is composed of 6 subsets (Alveolar macrophages, Monocytes, Neutrophils, Intermediate phase ARD, Late phase ARD and ARD cytokine storm). Proteins are listed by their UniProtKB identifier and under Effect the type of action of the protein is described: 1 (Being the protein more active contributes to the pathological effect of the motive) 0 (Being the protein less active contributes to the pathological effect of the motive).
(XLSX)

**S2 Table. Drug targets of pirfenidone and melatonin.** Drug targets are listed by their UniProtKB identifier and the pharmacological action indicates if the target is activated (+1) or inhibited (-1) by the drug.
(XLSX)

**S3 Table. Results of the GUILDify v2.0 web server analysis.** Sheet (a) summarizes the overlap between the top-scoring proteins associated with the drugs and the top-scoring proteins associated with SARS-CoV-2 infection sets. Sheet (b) contains figures that illustrate the overlap between top-scoring proteins associated with drugs and with SARS-CoV-2 infection sets. Sheet (c) contains the Gene Ontology enriched functions for the top-scoring proteins from both the drugs and the SARS-CoV-2 entry points. Sheets (d) to (k) contain the scores of the top-scoring proteins associated with the drugs and the host-key points. Sheets (l) to (q) contain the seeds associated with the drugs and the host-key points.
(XLSX)

## Acknowledgments

The authors acknowledge Dr. Pablo Coto from the Hospital Vital Alvarez-Buylla (Asturias, Spain) for his collaboration about details about the COVID-19 infection and specially the SARS molecular characterization, and Dr. Esther Ramírez for her contribution in the clinical trial design.

## Author Contributions

**Conceptualization:** Laura Artigas, Mireia Coma, Juan de la Haba-Rodriguez, Alex Olvera, Jose Barbera, Baldo Oliva, Jose Manuel Mas.

**Data curation:** Laura Artigas, Pedro Matos-Filipe, Joaquim Aguirre-Plans, Raquel Valls, Narcis Fernandez-Fuentes.

**Formal analysis:** Laura Artigas, Pedro Matos-Filipe, Joaquim Aguirre-Plans, Narcis Fernandez-Fuentes, Baldo Oliva, Jose Manuel Mas.

**Funding acquisition:** Narcis Fernandez-Fuentes, Baldo Oliva, Jose Manuel Mas.

**Investigation:** Laura Artigas, Mireia Coma, Pedro Matos-Filipe, Joaquim Aguirre-Plans, Judith Farrés, Raquel Valls, Narcis Fernandez-Fuentes, Juan de la Haba-Rodriguez, Alex Olvera, Baldo Oliva, Jose Manuel Mas.

**Methodology:** Laura Artigas, Pedro Matos-Filipe, Joaquim Aguirre-Plans, Raquel Valls, Narcis Fernandez-Fuentes, Baldo Oliva, Jose Manuel Mas.

**Project administration:** Judith Farrés, Jose Manuel Mas.

**Resources:** Narcis Fernandez-Fuentes, Juan de la Haba-Rodriguez, Alex Olvera, Jose Barbera, Rafael Morales, Baldo Oliva, Jose Manuel Mas.

**Software:** Laura Artigas, Pedro Matos-Filipe, Joaquim Aguirre-Plans, Raquel Valls, Narcis Fernandez-Fuentes, Jose Manuel Mas.

**Supervision:** Narcis Fernandez-Fuentes, Juan de la Haba-Rodriguez, Jose Barbera, Rafael Morales, Baldo Oliva, Jose Manuel Mas.

**Validation:** Baldo Oliva, Jose Manuel Mas.

**Visualization:** Laura Artigas, Mireia Coma, Pedro Matos-Filipe, Joaquim Aguirre-Plans, Judith Farrés, Raquel Valls, Narcis Fernandez-Fuentes, Alex Olvera, Baldo Oliva, Jose Manuel Mas.

**Writing – original draft:** Laura Artigas, Mireia Coma, Joaquim Aguirre-Plans, Judith Farrés, Raquel Valls, Narcis Fernandez-Fuentes, Alex Olvera, Baldo Oliva, Jose Manuel Mas.

**Writing – review & editing:** Laura Artigas, Mireia Coma, Joaquim Aguirre-Plans, Judith Farrés, Raquel Valls, Narcis Fernandez-Fuentes, Juan de la Haba-Rodriguez, Alex Olvera, Jose Barbera, Rafael Morales, Baldo Oliva, Jose Manuel Mas.

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
