## [Decision Letter · Decision Letter 0]

1 Jul 2020

PONE-D-20-13552

In-silico drug repurposing study predicts the combination of pirfenidone and melatonin as a promising candidate therapy to reduce SARS-CoV-2 infection progression and respiratory distress caused by cytokine storm

PLOS ONE

Dear Dr. Oliva,

Thank you for submitting your manuscript to PLOS ONE. After careful consideration, we feel that it has merit but does not fully meet PLOS ONE’s publication criteria as it currently stands. Therefore, we invite you to submit a revised version of the manuscript that addresses the points raised during the review process.

Both the reviewers are generally appreciative of the work. However Reviewer 1 notes major problems with presentation. There are also relatively minor suggestions from both the reviewers. We agree with the reviewers. We request you to submit a revised manuscript which addresses the criticisms and comments of both the reviewers.

We look forward to receiving your revised manuscript.

Kind regards,

Narayanaswamy Srinivasan

Academic Editor

PLOS ONE

Journal Requirements:

2.Thank you for stating the following in the Acknowledgments Section of your manuscript:

[JAP, NFF and BO acknowledge support from the Spanish Ministry of Economy (MINECO)

[BIO2017-85329-R] [RYC-2015-17519]; Unidad de Excelencia María de Maeztu”, funded

by the Spanish Ministry of Economy [ref: MDM-2014-0370].PMF has received funding

from the European Union’s Horizon 2020 research and innovation programme under

the Marie Skłodowska-Curie grant agreement No 765912.]

 [Funding for publication is from Agència de Gestió d’Ajuts Universitaris I de Recerca de la Generalitat de Catalunya [2017 SGR 01020]]

3.Thank you for stating the following in the Competing Interests section:

[The authors have declared that no competing interests exist.]

We note that one or more of the authors are employed by a commercial company: Anaxomics Biotech

Additional Editor Comments (if provided):

I agree with the comments of both the reviewers. As Reviewer 1 noted the manuscript needs significant improvement before a possible publication. I request authors to address comments and suggestions of both the reviewers in a revised manuscript for further consideration.

Reviewers' comments:

Reviewer's Responses to Questions

**Comments to the Author**

1. Is the manuscript technically sound, and do the data support the conclusions?

Reviewer #1: Partly

Reviewer #2: Yes

2. Has the statistical analysis been performed appropriately and rigorously? 

Reviewer #1: N/A

Reviewer #2: N/A

3. Have the authors made all data underlying the findings in their manuscript fully available?

Reviewer #1: Yes

Reviewer #2: Yes

4. Is the manuscript presented in an intelligible fashion and written in standard English?

Reviewer #1: No

Reviewer #2: Yes

5. Review Comments to the Author

Reviewer #1: Comments to manuscript: PONE-D-20-13552

There is definitely a pressing need to identify repurpose-able drug candidates that might be effective against SARS-CoV-2. As important as it is to let relevant manuscripts be published as quickly as possible, the novelty of the study conducted, it’s rigor and quality of presentation is equally important in my opinion.

Artigas et al. have employed a computational approach to identify new uses of existing drugs in light of the current COVID-19 pandemic. A Therapeutic Performance Mapping System (TPMS), that models protein pathways underlying a drug/pathology to understand a clinical outcome or a phenotype, has been used to accomplish the same. A web-tool GUILDify is used to corroborate their findings.

Overall, the study is fair, but the manuscript lacks clarity and requires major revisions in content and English grammar.

Major revisions 1

1) Pirfenidone, an antifibrotic drug, is known to be effective in reducing the rate of lung damage by 50%. A study published on May 15, 2020, by George, P.M. et al. (https://www.thelancet.com/journals/lanres/article/PIIS2213-2600(20)30225-3/fulltext) discusses the potential of antifibrotic therapies, including pirfenidone and nintedanib for the treatment of COVID-19. Authors are requested to make note of this and cite the paper.

2) Pirfenidone and melatonin, individually, are under clinical evaluation for their use as a drug to treat COVID-19. The authors propose a combinatorial therapy with these two drugs. However, DrugBank suggests that the metabolism of melatonin can be decreased when combined with pirfenidone. Can the authors elaborate on the same? Is there a possibility that this drug-drug interaction leads to melatonin toxicity, affecting circadian rhythm and immune functions?

3) Line 45: What do authors mean by “additive downstream effect on human proteins”?

4) Line 166: Please explain “the truth table”

5) Line 185: Please elaborate on I2D and NetCombo scoring function

6) Line 202: It will be useful to describe the human proteins identified that potentially interact with SARS-CoV-2 aside from the generic tag “viral entry”.

7) All supplementary tables require legends; the data described in Table-S2 and S3 are not clear.

8) Line 311: The network overlaps between treatment and infection set are not clear, and the supplementary tables aren’t very helpful. The authors are requested to describe the overlaps with an additional table or a figure.

9) Line 333: What is the role of ITM2C? What is the relevance of identifying its interacting partners MTNR1B and furin?

10) Line 351: How is a “linker” defined? For instance, proteins like SRC and ASGR2 seem connected to nodes with high degree of connections, while A1BG and USP17L30 are connected to nodes with at most 2 connections. What is the significance? Why are linkers not described in the manuscript?

11) Figure 3: What are the GO terms of the connections illustrated in Figure 3? Can the authors comment on off-target effects of the two drugs?

12) Figure 4: Absolute values alongside percentages will be helpful. What are the proteins affected by both pirfenidone and melatonin? What are the proteins modulated by either of the drug alone? What is the physiological relevance to this finding?

13) Figures 2, 4 and 5 are not publication-ready. These need to be re-done properly.

Major revisions 2

The entire manuscript needs major revisions in use of English language. I have detailed some below. However, the authors are advised to rewrite the manuscript with proper usage of English grammar:

Line 40: "Fast approaches" can be replaced with "Cost and time efficient approaches”

Line 41- 44 needs revision in grammar.

Line 45: identified as “a” candidate

Line 48: “measured a positive effect” is misleading. Authors need to rephrase.

Line 50: “These evidences” instead of “All this evidence”

Line 54: “presence of virus and those patients who are at risk of developing severe pulmonary complications.”

Line 64: “caused due to” instead of “produced”

Line 64: "is affecting" must be replaced with "is an active infection, that has affected at least hundreds of thousands of individuals around the world”

Line 68: "patients evolve to pneumonia" must be replaced with "patients develop respiratory complications such as pneumonia”

Line 81: “cost-effective ratio” must be replaced with “cost-benefit ratio”

Line 136: The sentence needs rephrasing since TPMS has not been broadly applied. The authors probably mean that network and systems biology (on which TPMS is based) has been used to address clinical questions.

Line 158: “bases” must be replaced with “basis”

Figure 4: “moudlated” must be replaced with modulated.

The manuscript by Artigas et al. requires major revisions in content as well as in English language. The authors can consider resubmission once all the comments have been addressed. The manuscript is not acceptable in its present form.

Reviewer #2: The manuscript titled “In-silico drug repurposing study predicts the combination of pirfenidone and melatonin as a promising candidate therapy to reduce SARS-CoV-2 infection progression and respiratory distress caused by cytokine storm” is well-written and timely. The work deals with identifying possible combination of drugs (pirfenidone and melatonin) which could interfere with the host cell invasion of SARS-CoV-2 and thus prevent the progression of COVID-19. The authors used a system biology approach that integrates available knowledge to finally filter out high confidence associations. The work is important to address the ongoing pandemic. Such an approach could be applied for identifying drug combinations related to other pathways involved in SARS-CoV-2 infection or diseases. Following suggestions if considered would improve the quality of the manuscript.

(1) Discussion from the authors on application of their methodology to other pathways or disease areas would improve the significance of their work.

(2) Consideration of findings from an important recent publication “A SARS-CoV-2-Human Protein-Protein Interaction Map Reveals Drug Targets and Potential Drug-Repurposing” could enrich the work presented in the current manuscript.

(3) Certain parameters like “Nº common”, “Degree” mentioned in the result tables need explanation.

6. PLOS authors have the option to publish the peer review history of their article (what does this mean?). If published, this will include your full peer review and any attached files.

Reviewer #1: No

Reviewer #2: No

---

## [Author Response · Author response to Decision Letter 0]

10 Sep 2020

NOTE: Formatted answers are in the cover letter, including figures and tables

Reviewer comments

Reviewer #1:

There is definitely a pressing need to identify repurpose-able drug candidates that might be effective against SARS-CoV-2. As important as it is to let relevant manuscripts be published as quickly as possible, the novelty of the study conducted, it’s rigor and quality of presentation is equally important in my opinion.

Artigas et al. have employed a computational approach to identify new uses of existing drugs in light of the current COVID-19 pandemic. A Therapeutic Performance Mapping System (TPMS), that models protein pathways underlying a drug/pathology to understand a clinical outcome or a phenotype, has been used to accomplish the same. A web-tool GUILDify is used to corroborate their findings.

Overall, the study is fair, but the manuscript lacks clarity and requires major revisions in content and English grammar.

Major revisions 1

1) Pirfenidone, an antifibrotic drug, is known to be effective in reducing the rate of lung damage by 50%. A study published on May 15, 2020, by George, P.M. et al. (https://www.thelancet.com/journals/lanres/article/PIIS2213-2600(20)30225-3/fulltext) discusses the potential of antifibrotic therapies, including pirfenidone and nintedanib for the treatment of COVID-19. Authors are requested to make note of this and cite the paper.

As the reviewer indicates, we have mentioned in the main text the antifibrotic role of pirfenidone and its positive implications in the treatment of COVID-19, citing the recent study by George P.M. et al:

The anti-inflammatory effect of both treatments could potentially be useful to reduce the effects caused by the cytokine storm. Pirfenidone has also been described as antifibrotic, reducing the rate of lung damage by 50% in patients with idiopathic pulmonary fibrosis [68]. Antifibrotic therapy has been highlighted as a potential strategy to treat COVID-19 patients and prevent fibrosis after SARS-CoV-2 infection [69].

[...]

68. King TE, Bradford WZ, Castro-Bernardini S, Fagan EA, Glaspole I, Glassberg MK, et al. A phase 3 trial of pirfenidone in patients with idiopathic pulmonary fibrosis. N Engl J Med. 2014;370: 2083–2092. doi:10.1056/NEJMoa1402582

69. George PM, Wells AU, Jenkins RG. Pulmonary fibrosis and COVID-19: the potential role for antifibrotic therapy [Internet]. The Lancet Respiratory Medicine. Lancet Publishing Group; 2020. doi:10.1016/S2213-2600(20)30225-3

2) Pirfenidone and melatonin, individually, are under clinical evaluation for their use as a drug to treat COVID-19. The authors propose a combinatorial therapy with these two drugs. However, DrugBank suggests that the metabolism of melatonin can be decreased when combined with pirfenidone. Can the authors elaborate on the same? Is there a possibility that this drug-drug interaction leads to melatonin toxicity, affecting circadian rhythm and immune functions?

Melatonin and pirfenidone are known to have a common cytochrome (CYP1A2) involved in their metabolization. Both drugs are substrates for CYP1A2 thus it is possible that its metabolism is inhibited because of the competition between the drugs. It is advisable to lower the dosage of the drugs in the drug-cocktail because they remain longer in the organism than in monotherapy. In spite of this adjustment, the combination is deemed safe by the medical doctors assessing this study. A study on drug interactions with melatonin showed that only in the case of a strong CYP1A2 inhibitor (not a substrate as is the case for pirfenidone) the 6-melatonin hydroxylation was impaired at pharmacologically relevant concentrations, and likely to lead to clinical interactions (Potential drug interactions with melatonin, May 2014Physiology & Behavior 131:17–24, DOI: 10.1016/j.physbeh.2014.04.016). In the case of pirfenidone the metabolism of the drug has minor contributions from other CYP isoenzymes, thus minimizing the effect of the competition over CYP1A2 for metabolization.

A comment in this respect has been included in the text:

From a medical point of view both drugs are considered safe and can be combined with the current standard of care treatments for COVID-19. The pharmacokinetics of the combination therapy can be altered respect to the individual drugs due to their competition (substrate – substrate) over CYP1A2. This substrate competition can lead to the drugs remaining longer in the organism than in monotherapy, potentially increasing the known pharmacologic effects of the drugs that in turn could lead to dose-dependent adverse drug reactions. The design of any clinical trial to test this combination therapy has to take into consideration this possibility and compensate by reducing the dosage of the drugs compared to their regular use in monotherapy.

3) Line 45: What do authors mean by “additive downstream effect on human proteins”?

The models have identified different pathways how the drugs may affect the known human proteins with a role in SARS-CoV-2 infection and they do converge at different points. The text has been modified to clarify this point.

The drug combination of pirfenidone and melatonin has been identified as a candidate treatment that may contribute to reduce the virus infection. Starting from different drug targets the effect of the drugs converges on human proteins with a known role in SARS-CoV-2 infection cycle..

4) Line 166: Please explain “the truth table”

The truth table is defined as the table used to determine if a statement is true or false. Is the term we use to refer to a set of validated benchmarks that we use for training the models. We changed the term to make it more general understandable.

The resulting subnetwork of proteins with positive and negative values defines the MoA of the drug. Sampling methods are used to generate mathematical models with stratified ensembles that comply with a set of validated benchmarks [12].

5) Line 185: Please elaborate on I2D and NetCombo scoring function

We agree with the reviewer that the text was not clear. I2D refers to the human biological network selected in GUILDify web server to perform the analysis. NetCombo refers to the scoring function chosen to score the proteins according to their proximity in the network with the seeds (proteins associated with the infection of SARS-CoV-19 or related phenotypes). Accordingly, we have modified the text in the manuscript:

GUILDify v2.0 has been run for each drug, using their targets as seeds (by introducing the name of the drug in the search box). The same has been done for each set of proteins: the coronavirus-host interaction set, the entry points set (a subset of the coronavirus-host interaction set containing only 5 host proteins relevant in the infection of SARS-CoV-2), the lung cells infection set and the ARD set (by uploading a file with the gene symbols of the proteins). We included the entry points subset in the analysis to have a more specific neighborhood of the SARS-CoV-2 infection mechanism. There are two additional parameters to be chosen by the user in GUILDify web server: the human biological network and the scoring function. The human biological network used has been the protein-protein interactions network of I2D [33], as it is the more complete network in the web server (with 13,568 proteins and 224,675 interactions). The scoring function selected (used to score the proteins of the network according to the proximity with the seeds) has been NetCombo [34], an algorithm that combines the performance of three network-based prioritization algorithms: NetShort (considers the number of edges connected to phenotype-associated nodes), NetZcore (iteratively assesses the relevance of a node for a given phenotype by averaging the normalized scores of the neighbors) and NetScore (considers the multiple shortest paths from the source of information to the target). The top 1% scored proteins of each test have been selected to proceed with the analysis of each subnetwork. The overlap between the selected subnetworks has been used to calculate the number of common proteins between the top-scoring proteins associated with the drugs and the top-scoring proteins associated to the host-key points (results detailed in File S3).

6) Line 202: It will be useful to describe the human proteins identified that potentially interact with SARS-CoV-2 aside from the generic tag “viral entry”.

In the same paragraph some of the human proteins are listed, but the term viral entry has been removed to avoid confusion as the rest of the paragraph describes the content of the set.

.

1) ‘Coronavirus-host interaction set’ is composed of human proteins with a relevant role in SARS-CoV-2 infection process. This set of proteins has been obtained from a bibliographic revision and it is composed of proteins that potentially interact with SARS-CoV-2: TMPRSS2 [37], ACE2 [37], GRP78 [38], Furin [39], CD147 [40]. In addition, a list of proteins reported to have a direct interaction with coronaviruses, retrieved from Zhou et al. [6] has been included. In this study, the authors define a human coronavirus (HCoV)-host interactome network including 119 host proteins with the aim of identifying antiviral drugs through repurposing strategies. It integrates known human direct targets of HCoV proteins or that are involved in crucial pathways of HCoV infection using the available information from four known HCoVs (SARS-CoV, MERS-CoV, HCoV-229E, and HCoV-NL63), one mouse CoV (MHV), and one avian CoV (IBV, N protein). The final set contains 122 human proteins, the full list is provided in Table S1.

7) All supplementary tables require legends; the data described in Table-S2 and S3 are not clear.

As suggested, we included more explicit legends for the supplementary tables both in the section “Supporting information” of the main text and inside the files:

 Supporting information

The following supplementary data is available online:

S1 Table: Characterization of the different sets of proteins associated to COVID-19.

1) coronavirus-host interaction set (including SARS-CoV-2 entry points), 2) lung-cells infection set, and 3) acute respiratory distress (ARD) set that is composed of 6 subsets (Alveolar macrophages, Monocytes, Neutrophils, Intermediate phase ARD, Late phase ARD and ARD cytokine storm). Proteins are listed by their UniProtKB identifier and under Effect the type of action of the protein is described: 1 (Being the protein more active contributes to the pathological effect of the motive) 0 (Being the protein less active contributes to to the pathological effect of the motive)

S2 Table: Drug targets of pirfenidone and melatonin. Drug targets are listed by their UniProtKB identifier and the action indicates if the target is activated (+1) or inhibited (-1) by the drug.

S3 File: Results of the GUILDify v2.0 web server analysis. Sheet (a) summarizes the overlap between the top-scoring proteins associated with the drugs and the top-scoring proteins associated with SARS-CoV-2 infection sets. Sheet (b) contains figures that illustrate the overlap between top-scoring proteins associated with drugs and with SARS-CoV-2 infection sets. Sheet (c) contains the Gene Ontology enriched functions for the top-scoring proteins from both the drugs and the SARS-CoV-2 entry points. Sheets (d) to (k) contain the scores of the top-scoring proteins associated with the drugs and the host-key points. Sheets (l) to (q) contain the seeds associated with the drugs and the host-key points.

The changes in S1 Table are the following:

The changes in S2 Table are the following:

The changes in S3 File are the following:

8) Line 311: The network overlaps between treatment and infection set are not clear, and the supplementary tables aren’t very helpful. The authors are requested to describe the overlaps with an additional table or a figure.

We have included three figures in the S3 File of Supplementary Data where we show the overlaps between treatment and the infection sets as Venn diagrams. In this way, we think that it is easier to visualize which sets of proteins are related with each other:

S3 Figure (b.1): Overlap between the top-scoring proteins associated with Pirfenidone targets and the sets of infection of the virus

S3 Figure (b.2): Overlap between the top-scoring proteins associated with Melatonin targets and the sets of infection of the virus

S3 Figure (b.3): Overlap between the top-scoring proteins associated with the targets of Pirfenidone and Melatonin combination and the sets of infection of the virus

9) Line 333: What is the role of ITM2C? What is the relevance of identifying its interacting partners MTNR1B and furin?

ITM2C is a transmembrane protein whose importance relies on the fact that it interacts with furin (one of the SARS-CoV-2 entry points and pirfenidone target) and also with MTNR1B (melatonin target). Therefore, the effect of melatonin on furin is mainly based on the interaction of MTNR1B with ITM2C and ITM2C with furin. By identifying the interacting partners of MTNR1A and MTNR1B, we can understand better the mechanism of action of the drug towards the SARS-CoV-2 entry points. We modified the main text so that this is clearer:

The highly significant overlap between pirfenidone and the top-scoring proteins of the 5 entry point proteins of SARS-CoV-2 is however biased, because furin is in this set and it’s also a target of pirfenidone (Fig 3). The therapeutic effect of pirfenidone on furin has a clear source effect on the neighbourhood of proteins that surround the entry points of SARS-CoV-2. Melatonin has also a significant overlap with the entry points of SARS-CoV-2. This effect is explained because it targets the proteins MTNR1A and MTNR1B (melatonin receptors). Melatonin affects SARS-CoV-2 GRP-78 entry point (also called HSPA5) because MTNR1A and MTNR1B are direct interactors of GRP-78 (Fig 3). Melatonin also affects SARS-CoV-2 furin entry point because MTNR1B interacts with the membrane protein ITM2C, which directly interacts with furin. By identifying the interacting partners of MTNR1A and MTNR1B, we can understand better the potential mechanism of action of melatonin towards the entry of the virus. 

10) Line 351: How is a “linker” defined? For instance, proteins like SRC and ASGR2 seem connected to nodes with high degree of connections, while A1BG and USP17L30 are connected to nodes with at most 2 connections. What is the significance? Why are linkers not described in the manuscript?

We define “linker” in the legend of Figure 3 as “proteins that link unconnected groups of proteins to the largest connected component”, though we agree that it is difficult to understand what a linker is from this definition. This is why we have tried to clarify it.

In Figure 3, we show how the top-scoring proteins from the melatonin, pirfenidone and entry points sets are connected between themselves. Still, there could be top-scoring proteins that might not be connected with other top-scoring proteins (e.g. MROH5 or PDILT), or groups of proteins that are disconnected from the largest connected component (e.g. the group of proteins surrounding ACE2). To connect these proteins or groups of proteins, we search for “linkers”: non-top-scoring proteins that can connect top-scoring proteins to the largest connected component. If there is more than one possible linker, we choose the one with the highest score. The function of the linker is mainly visual, for being able to understand better how the top-scoring proteins interact between themselves. But they are not relevant for the analysis. This is why we only described them in the legend of Figure 3 and not in the main text. 

In order to improve the comprehension of the article, we have improved the definition of linker from the legend of Figure 3:

Fig 3. Network of the SARS-CoV-2 entry points and the shared interactions with pirfenidone and melatonin. The extended set of proteins associated with the SARS-CoV-2 entry points, pirfenidone and melatonin were obtained with GUILDify v2.0. In the network, seeds are represented as octagons, non-seeds as circles and “linkers” as diamonds. “Linkers” are non-top-scoring proteins (they do not belong to any of the represented sets) that link unconnected groups of proteins to the largest connected component. “Linkers” are represented in this plot to facilitate the visualization, but they are not associated to any of the sets of proteins, thus having a less relevant role in the interpretation of the results. Pirfenidone-associated proteins are coloured in orange, melatonin-associated proteins are coloured in green and proteins associated with both drugs in brown. 

11) Figure 3: What are the GO terms of the connections illustrated in Figure 3? Can the authors comment on off-target effects of the two drugs?

 We have calculated the GO terms enriched for the proteins associated with both the entry points of SARS-CoV-2 and the treatments. We obtained 15 significantly enriched terms targeted either by pirfenidone or melatonin. Among them, we find functions such as “beta-amyloid binding”, “peptide binding” or “amide binding” that may involve potential off-targets (i.e. any of the nodes of the subnetworks affected). The tables of the functions are in S3 File of Supplementary Data:

S3 Table (c.1): Gene Ontology functions enriched for the top-scoring proteins from both the SARS-CoV-2 entry points and the Melatonin/Pirfenidone combination. Functional enrichment calculated with FuncAssociate 2.0.

# of genes # of total genes Log of odds ratio P-value Adjusted p-value GO term ID Go term name

4 52 1.67584 0 0.0145 GO:0001540 beta-amyloid binding

6 193 1.27013 0 0.0035 GO:0042277 peptide binding

6 221 1.20901 0.00001 0.0185 GO:0033218 amide binding

6 233 1.18518 0.00001 0.021 GO:0005578 proteinaceous extracellular matrix

7 357 1.0698 0.00001 0.021 GO:0004175 endopeptidase activity

11 867 0.91194 0 0.0005 GO:0044431 Golgi apparatus part

11 1019 0.83716 0.00001 0.018 GO:0005615 extracellular space

11 1167 0.77393 0.00002 0.045 GO:0031974 membrane-enclosed lumen

14 1714 0.74138 0.00001 0.018 GO:0005576 extracellular region

20 3563 0.63793 0.00001 0.0205 GO:0044421 extracellular region part

S3 Table (c.2): Gene Ontology functions enriched for the top-scoring proteins from both the SARS-CoV-2 entry points and Pirfenidone. Functional enrichment calculated with FuncAssociate 2.0.

# of genes # of total genes Log of odds ratio P-value Adjusted p-value GO term ID Go term name

2 4 2.77557 0.00002 0.0435 GO:0032190 acrosin binding

2 4 2.77557 0.00002 0.0435 GO:2000360 negative regulation of binding of sperm to zona pellucida

4 52 1.77139 0 0.0025 GO:0001540 beta-amyloid binding

5 193 1.28112 0.00002 0.0455 GO:0042277 peptide binding

6 312 1.15579 0.00001 0.028 GO:0030198 extracellular matrix organization

6 313 1.15435 0.00002 0.0285 GO:0043062 extracellular structure organization

9 867 0.91185 0.00001 0.0255 GO:0044431 Golgi apparatus part

13 1714 0.82675 0 0.0025 GO:0005576 extracellular region

18 3563 0.72192 0 0.0125 GO:0044421 extracellular region part

S3 Table (c.3): Gene Ontology functions enriched for the top-scoring proteins from both the SARS-CoV-2 entry points and Melatonin. Functional enrichment calculated with FuncAssociate 2.0.

# of genes # of total genes Log of odds ratio P-value Adjusted p-value GO term ID Go term name

3 107 1.90782 0.00002 0.0385 GO:0030168 platelet activation

12) Figure 4: Absolute values alongside percentages will be helpful. What are the proteins affected by both pirfenidone and melatonin? What are the proteins modulated by either of the drug alone? What is the physiological relevance to this finding?

We have included absolute values in figure 4 as indicated. References to the figure in text have been updated to reflect the change.

Fig 4. Proteins associated to ARD set modulated by pirfenidone and melatonin individually and in combination. The MoA has been calculated for each drug individually and in combination for each of the subsets defining ADR set. The first 3 columns provide the % and absolute value of affected proteins in the set for each MoA. Last column provides the cumulative, showing the protein overlaps affected between the different treatments.

The detail of the proteins predicted to be affected by each drug has been included in supplementary material table S1. Reference to this additional material has been included.

Figure 4 displays the % and absolute value of affected proteins predicted by the MoA by the individual drugs and by the drug combination for each of the ARD subsets. Last column provides the overlap between the proteins affected in each of the individual MoAs. The detail of the proteins affected is listed alongside the list of Set 3 proteins in Supplementary material Tables S1.

Screen shot of the modified Table S1.

In addition the following section in the manuscript “Mechanism of action of pirfenidone-melatonin combination over respiratory distress associated proteins” details the proteins that appear the most represented in the MoA distribution and their role is discussed.

13) Figures 2, 4 and 5 are not publication-ready. These need to be re-done properly.

New Figures 2, 4, 5, are provided.

Major revisions 2

The entire manuscript needs major revisions in use of English language. I have detailed some below. However, the authors are advised to rewrite the manuscript with proper usage of English grammar:

We would like to thank the reviewer for the valuable corrections. We have corrected all them, and we have had the manuscript proofread by native English professionals.

Line 40: "Fast approaches" can be replaced with "Cost and time efficient approaches”

Cost and time efficient approaches to reduce the burthen of the disease are needed.

Line 41- 44 needs revision in grammar.

To find potential COVID-19 treatments among the whole arsenal of existing drugs, we combined system biology and artificial intelligence-based approaches.

Line 45: identified as “a” candidate

The drug combination of pirfenidone and melatonin has been identified as a candidate with an additive downstream effect on human proteins that may contribute to reduce the virus infection.

Line 48: “measured a positive effect” is misleading. Authors need to rephrase.

We have also predicted a potential therapeutic effect of the drug combination over the respiratory associated pathology, thus tackling at the same time two important issues in COVID-19.

Line 50: “These evidences” instead of “All this evidence”

Line 54: “presence of virus and those patients who are at risk of developing severe pulmonary complications.”

These evidences, together with the fact that from a medical point of view both drugs are considered safe and can be combined with the current standard of care treatments for COVID-19 makes this combination very attractive for treating patients at stage II, non-severe symptomatic patients with the presence of virus and those patients who are at risk of developing severe pulmonary complications.

Line 64: “caused due to” instead of “produced”

Line 64: "is affecting" must be replaced with "is an active infection, that has affected at least hundreds of thousands of individuals around the world”

The pandemic disease COVID-19 caused due to SARS-CoV-2 is an active infection, that has affected at least hundreds of thousands of individuals around the world.

Line 68: "patients evolve to pneumonia" must be replaced with "patients develop respiratory complications such as pneumonia”

Within a week 20% of the patients develop respiratory complications such as pneumonia, with chest tightness, chest pain, and shortness of breath [2].

Line 81: “cost-effective ratio” must be replaced with “cost-benefit ratio”

Drug repurposing (i.e. reusing existing drugs approved for different conditions and with acceptable safety profiles) is emerging as a rapid and excellent cost-benefit ratio alternative to find treatments against SARS-CoV-2, while vaccines and new treatments are being developed.

Line 136: The sentence needs rephrasing since TPMS has not been broadly applied. The authors probably mean that network and systems biology (on which TPMS is based) has been used to address clinical questions.

The TPMS technology employed has been previously described [10] and broadly applied in different clinical areas with different objectives [5,6,11–16].

Line 158: “bases” must be replaced with “basis”

On the basis of the human protein functional network described above, mathematical models have been built.

Figure 4: “moudlated” must be replaced with modulated.

Fig 4. Percentage of proteins associated to ARD set modulated by pirfenidone and melatonin individually and in combination.

The manuscript by Artigas et al. requires major revisions in content as well as in English language. The authors can consider resubmission once all the comments have been addressed. The manuscript is not acceptable in its present form.

 

Reviewer #2: 

The manuscript titled “In-silico drug repurposing study predicts the combination of pirfenidone and melatonin as a promising candidate therapy to reduce SARS-CoV-2 infection progression and respiratory distress caused by cytokine storm” is well-written and timely. The work deals with identifying possible combination of drugs (pirfenidone and melatonin) which could interfere with the host cell invasion of SARS-CoV-2 and thus prevent the progression of COVID-19. The authors used a system biology approach that integrates available knowledge to finally filter out high confidence associations. The work is important to address the ongoing pandemic. Such an approach could be applied for identifying drug combinations related to other pathways involved in SARS-CoV-2 infection or diseases. Following suggestions if considered would improve the quality of the manuscript.

(1) Discussion from the authors on application of their methodology to other pathways or disease areas would improve the significance of their work.

We have extended the references of the successful application of TPMS for drug repurposing. 

The TPMS technology is a top-down approach which mathematically models the human physiology by integrating the known information about the functional interaction of proteins with experimental data and patient data available in public repositories, with a focus on clinical translation. There are already several examples of repurposed drugs identified using TPMS approach that have been validated in vitro and in vivo [8,9,12–18] with one of them advancing to clinical trials.

(2) Consideration of findings from an important recent publication “A SARS-CoV-2-Human Protein-Protein Interaction Map Reveals Drug Targets and Potential Drug-Repurposing” could enrich the work presented in the current manuscript.

Although we considered the work of Gordon et al., we did not used their proposed interactions because there were some important proteins for the infection of the virus that were missing (e.g. ACE2 or furin). Still, we do think that the work is very relevant to understand the context of systems biology research on COVID-19. Thus, we mentioned their work in the main text:

However, there are two key aspects to be considered when searching for drug repurposing candidates. First of all, identify the key proteins or pathways to be targeted by the treatment. These target elements can either be the proteins involved in the mechanisms used by the virus to enter the host cell and replicate, or the host proteins that facilitate the spread of the virus or cause over-reactive dangerous responses. Proteomics studies such as the works of Gordon et al. [5], Zhou et al. [6] or IntAct [7] are uncovering the target elements of the human interactome.

[...]

5. Gordon DE, Jang GM, Bouhaddou M, Xu J, Obernier K, White KM, et al. A SARS-CoV-2 protein interaction map reveals targets for drug repurposing. Nature. 2020;583: 459–468. doi:10.1038/s41586-020-2286-9

(3) Certain parameters like “Nº common”, “Degree” mentioned in the result tables need explanation.

“Nº common” refers to the number of common proteins between the pairs of sets analyzed in the table. We have replaced “Nº common” by “Nº common proteins” in S3 Table (a), and we have made a legend for the table where we clarify the meaning of “Nº common proteins” and “P-value”: 

The “degree” of a node (in graph theory) is the number of edges of the node. In our case, it is the number of interactions that a protein has with other proteins of the network. We have made legends for S3 Tables (b) to (i) where we clarify the meaning of “Degree”:

---

## [Editor Report · Decision Letter 1]

22 Sep 2020

In-silico drug repurposing study predicts the combination of pirfenidone and melatonin as a promising candidate therapy to reduce SARS-CoV-2 infection progression and respiratory distress caused by cytokine storm

PONE-D-20-13552R1

Dear Dr. Oliva,

We’re pleased to inform you that your manuscript has been judged scientifically suitable for publication and will be formally accepted for publication once it meets all outstanding technical requirements.

Kind regards,

Narayanaswamy Srinivasan

Academic Editor

PLOS ONE
---

## [Editor Report · Acceptance letter]

24 Sep 2020

PONE-D-20-13552R1 

In-silico drug repurposing study predicts the combination of pirfenidone and melatonin as a promising candidate therapy to reduce SARS-CoV-2 infection progression and respiratory distress caused by cytokine storm 

Dear Dr. Oliva:

I'm pleased to inform you that your manuscript has been deemed suitable for publication in PLOS ONE. Congratulations! Your manuscript is now with our production department. 

Kind regards, 

on behalf of

Prof. Narayanaswamy Srinivasan 

Academic Editor

PLOS ONE